# An Exploratory Analysis on the 2D:4D Digit Ratio and Its Relationship with Social Responsiveness in Adults with Prader–Willi Syndrome

**DOI:** 10.3390/jcm12031155

**Published:** 2023-02-01

**Authors:** Sara Gámez, Jesus Cobo, Meritxell Fernández-Lafitte, Ramón Coronas, Isabel Parra, Joan Carles Oliva, Aida Àlvarez, Susanna Esteba-Castillo, Olga Giménez-Palop, Raquel Corripio, Diego J. Palao, Assumpta Caixàs

**Affiliations:** 1Mental Health Department, Corporació Sanitària Parc Taulí—Universitat Autònoma de Barcelona—CIBERSAM, 08202 Sabadell, Spain; 2Department of Psychiatry and Forensic Medicine, Universitat Autònoma de Barcelona, 08193 Bellaterra, Spain; 3Institut d’Investigació i Innovació Parc Taulí (I3PT)—CERCA, 08208 Sabadell, Spain; 4Statistics Unit, Fundació Parc Taulí—(I3PT)—CERCA, 08208 Sabadell, Spain; 5Department of Mental Health, Mutua Terrassa University Hospital, 08221 Terrassa, Spain; 6Specialized Mental Health and Intellectual Disability Department, Institut d’Assistència Sanitària, Parc Hospitalari Martí i Julià, 17190 Girona, Spain; 7Neurodevelopment Group, Girona Biomedical Research Institute IDIBGI, Institut d’Assistència Sanitària, Parc Hospitalari Martí i Julià, 17190 Girona, Spain; 8Endocrinology and Nutrition Department, Hospital Universitari Parc Taulí, Corporació Sanitària Parc Taulí—Universitat Autònoma de Barcelona, 08202 Sabadell, Spain; 9Department of Medicine, Universitat Autònoma de Barcelona, 08193 Bellaterra, Spain; 10Pediatric Endocrine Department, Parc Taulí Hospital Universitari, Institutd’Investigació i Innovació Parc Taulí (I3PT), Universitat Autònoma de Barcelona, 08202 Sabadell, Spain

**Keywords:** Prader–Willi syndrome, epigenetic, testosterone, estradiol, prenatal, D2:D4, social responsiveness, social functioning, functionality, function

## Abstract

Prader–Willi syndrome (PWS) is a genetic disorder produced by a lack of expression of paternally derived genes in the 15q11–13 region. Research has generally focused on its genetic and behavioral expression, but only a few studies have examined epigenetic influences. Prenatal testosterone or the maternal testosterone-to-estradiol ratio (MaTtEr) has been suggested to play an important role in the development of the ‘social brain’ during pregnancy. Some studies propose the 2D:4D digit ratio of the hand as an indirect MaTtEr measure. The relationship between social performance and MaTtEr has been studied in other neurodevelopmental conditions such as Autism Spectrum Disorder (ASD), but to our best knowledge, it has never been studied in PWS. Therefore, our study aims to clarify the possible existence of a relationship between social performance—as measured using the Social Responsiveness Scale (SRS)—and MaTtEr levels using the 2D:4D ratio. We found that, as a group, PWS individuals have shorter index and ring fingers than the control group, but no significant difference in the 2D:4D ratios. The 2D:4D ratio showed a correlation only with Restricted Interests and Repetitive Behavior Subscale, where a positive correlation only for male individuals with PWS was found. Considering only PWS with previous GH treatment during childhood/adolescence (PWS-GH), index and ring fingers did not show differences in length with the control group, but the 2D:4D ratio was significantly higher in the right or dominant hand compared to controls.

## 1. Introduction

Prader–Willi syndrome (PWS) is both a genetic and epigenetic disorder, mapping the imprinted chromosomal domain of 15q11.2–13.3. This critical region includes different genes, and it has been described that all cases of PWS have an absence of an expressed paternal copy of the SNORD116 locus [1]. Moreover, due to parental imprinting of the locus, loss of the SNORD116 function can occur through deletion, uniparental disomy, or imprinting error [1]. Most cases result from deletion (65–75%) or maternal uniparental disomy (20–30%), and a few (1–3%) result from rare imprinting defects [2]. In fact, PWS is associated with epigenetic modifications with differences in SNORD116 and MAGEL2 mutations, which seem to be relevant to the different associated phenotypes [3].

Its prevalence is very low, ranging between 1:10,000 and 1:30,000 births. These genotypes (and some phenocopies) result in a complex phenotype characterized by hypotonia, hyperphagia (with a gradual development of morbid obesity unless eating is externally controlled), and hypogonadism. Morphological alterations, such as characteristic facial features and short stature, are described during the development of the body [4,5]. In addition, people with PWS often manifest psychopathological traits of compulsivity, rigidity, irritability, and social dysfunction [2,6]. Cognitive capacity in PWS usually ranges from borderline to moderate intellectual disability [6,7].

In addition to somatic and behavioral features, comorbidities have also been described in up to 89% of patients with PWS. The most common psychiatric comorbidities in PWS are affective disorders, psychosis, obsessive-compulsive disorder, and autism spectrum disorder (ASD) [8]. Some of the clinical characteristics of PWS differ according to the specific genetic abnormality [9,10,11,12]. Various previous studies have analyzed the prevalence and relevance of symptoms of ASD in individuals with PWS [4,8,12,13,14], finding that some of the mutations identified in segment 15q11–q13 are present in both PWS and ASD, which may indicate a common genetic background [15]. Individuals with PWS and ASD, unlike those without ASD, have lower IQ scores, worse social and verbal abilities, more stereotyped behaviors, and more restricted interests [8,12].

Concerning our own PWS populations, the rate of impaired social responsiveness (a core symptom of ASD) was identified by Fernández-Laffite et al. [16] in 76.9% of participants, and moderate to marked difficulties in social functioning were identified in 50%. Participants with impaired social responsiveness had significantly worse scores in functionality. Moreover, scores for the Social Cognition domain of the SRS scale positively correlated with the Socially useful activities (*p* < 0.05) and Personal and social relationships (*p* < 0.01) main areas of the PSP functionality scale. These results suggest that difficulties in social skills should be assessed in all psychosocial evaluations of patients with PWS, as well as ASD symptoms.

The 2D:4D digit ratio (the length of the second digit divided by the length of the fourth digit) has been proposed as a retrospective marker of prenatal maternal testosterone (PT) and testosterone-to-estradiol ratio (MaTtEr), with higher concentrations of testosterone or higher ratios of testosterone-to-estradiol during prenatal development resulting in lower 2D:4D ratios [17,18,19]. Some studies have found sex differences in the 2D:4D ratio in the right hand, with males obtaining smaller values than women [20,21,22]. Furthermore, the 2D:4D ratio at 2 years of age was found to correlate with PT and estradiol ratios during pregnancy [17,23]. Significantly, Hönekopp et al. [20] reported in a meta-analysis that individuals with excessive PT levels caused by congenital adrenal hyperplasia (CAH) a have lower 2D:4D ratio than unaffected sex-matched controls. In addition, some studies have suggested a relationship between elevated prenatal testosterone (PT) and the presence of autistic traits, such as difficulties in social cognition [24,25]. This suggests that PT has organizational effects on the brain and behavior [26] and may shape the neural mechanisms underlying social development.

Specific anthropometric features in PWS were determined decades ago. It was proposed that physical characteristics such as small hands and feet (acromicria) are typical manifestations of PWS, although there is not a universal consensus on this fact [27,28]. In Butler’s review including 538 patients, short stature and small hands and feet were reported in 76% and 83% of the patients, respectively [27]. Additionally, they found 57 subjects with PWS with acromicria or small hand and foot size [27]. Similarly, in a study with 56 cases of PWS, Hudgins and Cassidy found foot length to be proportionately smaller than hand length in all individuals [28]. Interestingly, this difference was more striking in females: At age 12, hand length for females was below the 25th centile and, in almost all cases, height was below the 50th centile. In contrast, hand length data for males appeared to be more within the normal range until adulthood. There were some effects attributable to racial differences, as Black individuals with PWS in their sampler had relatively larger hands and feet than their Caucasian counterparts [28]. Butler et al., in their previously mentioned study, found a significant negative correlation with age. However, no significant differences in anthropometric data were found between the sexes in individuals less than 10 years of age [27]. Moreover, no differences were found in the anthropometric measurements between the different genetic PWS subgroups. These initial measurements only included 3D middle finger length (not 2D or 4D fingers) [27,28,29].

Growth Hormone (GH) treatments in pediatric and adults with PWS have proven to improve body morphology and composition, physical performance, cognition, psychomotor development, respiratory function, and quality of life with few adverse effects [30,31,32] and could influence anthropometric measures. In the randomized controlled GH trial of Festen et al. [32], anthropometric parameters were assessed once every 3 months. They found that head circumference increased significantly to completely normal values during the GH treatment trial, whereas tibia length, foot length, arm span, and sitting height significantly improved but remained significantly lower [32]. Moix et al. [30] found that there is an increase in final height, normalization of cranial diameter, sitting/total height ratio, and improvement in SDS of hands, feet, tibial length, and arms in PWS following treatment with GH during the childhood/adolescence period [30].

To our best knowledge, there are not any previous studies focused on 2D:4D ratios in adult individuals with PWS nor in social responsiveness difficulties in PWS concerning levels of MaTtEr. Therefore, the purpose of this study was to describe characteristics and possible sex dysmorphisms in the 2D:4D ratios in adult individuals with PWS, compared to a control population, and to explore its relationship with social responsiveness and social functioning in the PWS sample. Based on previous results with other populations (mostly ASD populations), we hypothesized that people with PWS would have lower 2D:4D ratios than controls and that sex interactions could be different in the PWS sample than in controls. Moreover, we hypothesized that lower 2D.4D ratios will be associated with lower scores in social responsiveness and worse social functioning in the PWS population.

## 2. Materials and Methods

### 2.1. Design and Sample

This cross-sectional study included Caucasian adult patients with genetically diagnosed PWS who attended the endocrinology department in our reference Center for Rare Diseases at Consorci Corporació Sanitària Parc Taulí (Sabadell, Barcelona, Spain). Sixty-three adult participants took part in the present study, 27 with a formal diagnosis of PWS and 36 controls. The control group was composed of Caucasian university students and young clinical staff working at the same hospital.

### 2.2. Procedure

All participants with PWS and their legal guardians voluntarily agreed to participate in this study after being informed of the aim of it and signing the appropriate informed consent and assent.

To collect demographic and clinical data, including current treatments, several questionnaires were administered to the families and some anthropometric parameters, including index and ring fingers (2D and 4D fingers), were collected from the participants. All data were anonymized to preserve confidentiality.

### 2.3. Assessment

Intellectual disability and other psychiatric disorders were diagnosed according to the fifth edition of the Diagnostic and Statistical Manual of Mental Disorders (DSM-5) [33].

To measure autistic symptomatology and the severity of associated social impairment, we used the parent rating scale of the Spanish version of the Social Responsiveness Scale (SRS) [34], which assesses the presence and severity of social impairment within the autism spectrum in individuals from 2 years and 6 months of age through adulthood [35,36]. Its cross-cultural validity has been widely demonstrated [37]. The scale comprises 65 items scored on a Likert scale ranging from 1 (not true) to 4 (almost always true), with 17 items being reverse-scored. Higher SRS scores represent more ASD-related behaviors. The SRS evaluates five domains (Social Awareness, Social Cognition, Social Communication, Social Motivation, and Restricted Interests and Repetitive Behavior) that can be useful in clinical settings or for developing treatment plans. The domain of Social Awareness measures the ability to perceive social cues (e.g., “Is aware of what others are thinking or feeling”). The domain of Social Cognition measures the ability to interpret social cues once they are perceived (e.g., “Doesn’t recognize when others are trying to take advantage of him or her”). The domain of Social Communication measures expressive social communication (e.g., “Avoids eye contact or has unusual eye contact”). The domain of Social Motivation measures the extent to which a respondent is generally motivated to engage in social-interpersonal behavior, including elements of social anxiety, inhibition, and empathic orientation (e.g., “Would rather be alone than with others”). Finally, the domain of Restricted Interests and Repetitive Behavior measures stereotypical behaviors or highly restricted interests (e.g., “Has an unusually narrow range of interests”). SRS T-scores ≥ 60 indicate mild-to-moderate risk for ASD; this cut-off yields a 96.8% likelihood of a later clinical diagnosis of ASD [35]. The SRS’s ease of administration and strong psychometric properties have favored its widespread use in research [35,37] and for that reason was used in our previous analysis [16].

Behavioral and emotional current symptoms in the PWS sample were measured by the Strengths and Difficulties Questionnaire (SDQ) [38,39,40]. The SDQ is a quick and easy-to-perform questionnaire used in the screening of behavioral problems in childhood and adolescence. The use of this questionnaire was decided on the basis of the difficulties to find an optimal and adapted adult questionnaire valid for PWS and other genetic and neurodevelopmental disorders. This test was translated into Spanish, and it is widely used in Spanish epidemiological studies [41]. It comprises five subscales relevant to measuring internalizing and externalizing problems: Emotional Problems, Problems with Peers, Behavioral Problems, Hyperactivity, and Prosocial Behavior. It has been used in different adult samples in the field of disability diseases [42,43]. In our study, the Parent’s Version was used, and it was fulfilled by the main caregiver.

To evaluate patients’ social functionality, we used the Spanish version of the Personal and Social Performance scale (PSP) [44,45]. This clinician-rated instrument evaluates patients’ social functioning in four main areas of social and individual performance: Socially useful activities, Personal and social relationships, Self-care, and Disturbing and aggressive behaviors. We classified patients’ degree of difficulties in each main area in the following ranges: Absent; mild; manifest, but not marked; marked; severe; or very severe. Higher scores in the main areas indicate more severe difficulties.

To measure the severity of the PWS, we used a simple Visual-Analytical Severity Scale—VAS [46]. The scale used was a Likert scale, and it was administered by an experienced clinician (JC). Scores ranged between 0 and 10, where 0 represents no awareness or severity at all and 10 represents maximum awareness or severity of the illness.

There are different strategies to measure 2D:4D digit ratios [47]. In our study, the digit ratio measurement of each hand was obtained by the same researcher (JC) using the same procedure: A Caliper Vernier was used for the ventral measurement in the second finger (2D). The measurement was taken from the middle of the crease of the first phalanx to the middle of the ball of the finger (see Figure 1 and Figure 2). This procedure was repeated for the fourth finger (4D). Finally, the length of the second digit was divided by the length of the fourth digit to obtain a measure of the 2D:4D ratio for each hand separately [47,48]. We also included the measure of the dominant hand, which was identified by asking the volunteers. In case of any doubt, the dominant hand was identified as the writing hand or main hand for delicate manual activities.

Blood was extracted from all PWS volunteers at 8 a.m. after overnight fasting. A routine automated analyzer was used for laboratory tests (LH, FSH, Testosterone in males, and Estradiol in females).

### 2.4. Statistical Analysis

We used descriptive statistics, including counts, means, and standard deviations plus medians and interquartile range (ITQ, 25–75), when appropriate. To determine associations between variables, we used Spearman’s rank correlation coefficient or the intraclass correlation coefficient as appropriate. To compare variables between groups, we used non-parametric statistics (Mann–Whitney U or chi-square tests, as appropriate). Statistical significance was set at *p* < 0.05. We used IBM SPSS Statistics for Windows version 21 (IBM Corp., Armonk, NY, USA) for all analyses.

## 3. Results

### 3.1. Sociodemographic, Hormonal, and Anthropometric Variables in Both Groups

Descriptive statistics of basal sociodemographic variables for PWS and the control group are detailed in Table 1. Descriptive results by gender are detailed in Table 1a (for females) and Table 1b (for males).

[Fig jcm-12-01155-ch001] and [Fig jcm-12-01155-ch002] showed the bar graphs with raw data plotting related to 2D or 4D length and 2D:4D ratios, by group (PWS vs. controls) and by gender.

Fourteen PWS participants (51.9%) received GH administration during childhood/adolescence. We know the duration of the GH treatment in nine cases (Mean of 7.33 years, SD of 1.5; Median of 8.0 years, interquartile range of 6.0 to 8.0 years), during a range of ages from the first year of life to 17 years of age depending on the case. None of the PWS participants receive GH during adulthood after the age of 18.

Up to 17 PWS cases (63%) received current treatment for their hypogonadism: 9 male PWS participants received exogenous testosterone and 8 female PWS participants received exogenous low-dose estrogens.

Current FSH levels were significantly lower in PWS participants receiving treatment for their hypogonadism (U 46.0; *p* = 0.054). Current FH levels were not significantly affected by hypogonadism treatments. In female PWS, current estrogen levels were significantly affected (increased) by exogenous estrogen administration (U 11.5; *p* = 0.054). In male PWS, current testosterone levels were not significantly affected by exogenous testosterone treatments.

PWS participants have smaller index and ring fingers compared with our control group, but there are no significant differences in the 2D:4D ratio (Table 1a,b).

### 3.2. Bivariate Correlation of Severity VAS, Estimated Intelligence Quotient (IQ), Behavior (SDQ), Functionality (SPS), and Social Responsiveness (SRS) in the PWS Sample

Analysis using Spearman’s Rho found no correlation between estimated premorbid IQ and any 2D:4D digit ratio. Total VAS of Severity scores was also not significantly associated with any 2D:4D digit ratio (Table 2). No correlation of any 2D:4D ratio with any functionality measure (including total PSP scores or PSP subscales scores) was found in the total PWS sample. No correlation of any 2D:4D ratio with any SDQ measure (including total SDQ scores or SDQ subscales scores) was found in the total PWS sample. There were no correlations of any 2D:4D ratio with hormonal laboratory tests (LH, FSH, Testosterone in males, and Estradiol in females).

No correlation between any 2D:4D ratio with any SRS measure (including total SRS scores or SRS subscales scores) was found in the 26 PWS cases (Table 2).

### 3.3. Sex Interactions in Both Samples

Considering all participants in the PWS group, the Mann–Whitney U test showed no interaction between sex and the length of the index finger on the right hand (U = 86.5; Z = −0.171; *p* = 0.864 (two-tailed)), the length of the ring finger on the right hand (U = 84.5; Z = −0.269; *p* = 0.788 (two-tailed)), and the 2D:4D ratio of the right hand (U = 76.0; Z = −0.692, *p* = 0.489 (two-tailed)).

Likewise, in the PWS group, no interaction was found either between sex and the length of the index finger on the left hand (U = 85.0; Z = −0.245; *p* = 0.807 (two-tailed)), the length of the ring finger on the left hand (U = 79.5; Z = −0.513; *p* = 0.608 (two-tailed)), and the 2D:4D ratio on the left hand (U = 65.5; Z = −1.196; *p* = 0.232 (two-tailed)). No interaction was found between sex and the length of the index finger on the dominant hand (U = 86.5; Z = −0.171; *p* = 0.864 (two-tailed)), the length of the ring finger on the dominant hand (U = 85.0; Z = −0.244; *p* = 0.807 (two-tailed)), and the 2D:4D ratio on the dominant hand (U = 80.5; Z = −0.464; *p* = 0.642 (two-tailed)).

In the control group, the Mann–Whitney U showed an interaction between sex and the length of the index finger on the right hand (U = 59.5; Z = −2.850; *p* = 0.004 (two-tailed)) and the length of the ring finger on the right hand (U = 59.5; Z = −2.847; *p* = 0.004 (two-tailed)), but no interaction was found between the 2D:4D ratio on the right hand (U = 123.5; Z = −0.698; *p* = 0.485 (two-tailed)).

Likewise, in the control group, an interaction was found between sex and the length of the index finger on the left hand (U = 56.5; Z = −2.944; *p* = 0.003 (two-tailed)) and the length of the ring finger on the left hand (U = 48.5; Z = −3.212; *p* = 0.001 (two-tailed)) but no interaction was found with the 2D:4D ratio of the left hand (U = 133.5; Z = −0.354; *p* = 0.723 (two-tailed)). A marginal interaction was found between sex and the length of the index finger on the dominant hand (U = 63.0; Z = −2.733; *p* = 0.006 (two-tailed)) and the length of the ring finger on the dominant hand (U = 62.5; Z = −2.745; *p* = 0.006 (two-tailed)), but no association was found with the 2D:4D ratio of the dominant hand (U = 123.5; Z = −0.693; *p* = 0.488 (two-tailed)). Finally, the Mann–Whitney U test showed an interaction between sex and weight (U = 4.0; Z = −3.292; *p* < 0.001 (two-tailed)) and height (U = 20.5; Z = −4.151; *p* < 0.001 (two-tailed)), but not BMI (U = 90.0; Z = −1.812; *p* = 0.070 (two-tailed)).

### 3.4. Relationship and SRS Sex Interactions of 2D:4D Digits on the Right Hand in the PWS Sample

No correlation was found between the 2D:4D digit ratio of the right hand and the 2D:4D digit ratio of the left hand (*p* = 0.075; r= −0.348).

When controlling for sex, there was also no correlation between the 2D:4D digit ratio of the right hand and SRS Total Scores and Subscales Scores in women with PWS (Table 3).

SRS Restricted Interests and the Repetitive Behavior Subscale showed a significant relationship with the 2D:4D digit ratio on the right hand (*p* = 0.011; r = 0.728) in men with PWS (Table 4). There was no correlation between the 2D:4D digit ratio in the right hand and SRS Social Awareness Scale, SRS Social Cognition Scale, SRS Social Communication Scale, SRS Social Motivation Scale, and SRS Total Score in men with PWS (Table 4).

### 3.5. Relationship and SRS Sex Interactions of 2D:4D Digit on the Left Hand in the PWS Sample

When controlling for sex, there was no correlation between the 2D:4D digit ratio on the left hand and SRS Total Scores and Subscales Scores in women or in men with PWS (Table 3 and Table 4).

### 3.6. Relationship and SRS Sex Interactions of 2D:4D Digit on the Dominant Hand in the PWS Sample

When controlling for sex, there was no correlation between the 2D:4D digit ratio on the dominant hand and SRS Total Score or their Subscales in women or men with PWS (Table 3 and Table 4).

### 3.7. Relationship between 2D or 4D Finger Lengths, 2D:4D Digit Ratios, and GH Administration during the Childhood and Adolescent Period

PWS participants receiving treatment with GH during the childhood/adolescent (PWS-GH) period showed significant differences compared to PWS participants without treatment (PWS-NoGH) regarding the length of 2D and 4D, but no differences were found related to the 2D:4D ratio (Table 5).

PWS-GH participants showed some significant differences compared to healthy controls (Table 5). PWS-GH participants have no significant differences compared to controls in 2D or 4D length; however, they showed a significantly higher ratio compared to our healthy control sample in the 2D:4D ratio, in both the right hand and the dominant hand (Table 5).

[Fig jcm-12-01155-ch003] and [Fig jcm-12-01155-ch004] show bar graphs with raw data plotted related to 2D or 4D length and 2D:4D ratios, by GH treatment and by gender.

### 3.8. Relationship of GH Administration during the Childhood and Adolescent Period with Hormonal and SRS Factors

PWS-GH and PWS-NoGH groups did not show significant differences in any current hormonal measures, nor in any SRS measures.

## 4. Discussion

We hypothesized that people with PWS would have a lower 2D:4D ratio and higher difficulties in social responsiveness, which would be associated with worse psychosocial functioning. In our PWS sample, there are significant differences in 2D or 4D lengths related to our control group, but there are no significant differences with any 2D:4D ratios. Moreover, the global severity of the PWS was also not related to any 2D:4D digit ratio in our sample.

We could not find any relationship between the 2D:4D ratio in most domains of the SRS, which might indicate that the 2D:4D ratio is not a good predictor of social responsiveness for PWS. However, our results show a positive correlation between the SRS Restricted Interests and Repetitive Behavior Scale and the 2D:4D digit ratio of the right hand for males with PWS, but not for females. This could indicate that males tend to exhibit more restricted interests and repetitive behavior depending on their prenatal MaTtEr in the opposite direction of our initial hypothesis for the whole PWS sample, but the sample limitations of our exploratory analysis can only suggest these associations.

We also explored sex differences in 2D:4D ratios, as according to previous studies, it is expected to find differences between males and females. Our results show an interaction between sex and the length of the index finger on both hands and the length of the ring finger on both hands for the control group, with women having smaller fingers than men, in accordance with previous studies [20,21,22]. However, this finding does not apply to the PWS group, as no interaction between sex and the length of 2D or 4D fingers on any hand was found. These results might suggest that MaTtEr could affect PWS subjects more than control subjects. Similarly, we did not find an interaction between sex and 2D:4D ratios in the control group, as previous studies suggest [20,21,22], nor an interaction between sex and 2D:4D ratios in the PWS group.

As we cited in the introduction, GH treatments in pediatric and adult individuals with PWS have been proven to improve body morphology and composition, physical performance, cognition, psychomotor development, respiratory function, and quality of life [30,31,32]. In our sample, when we analyzed the data of the PWS participants considering their previous treatment with GH during childhood/adolescence, we found significant differences in the length of the index and ring fingers, but no differences were found on the 2D:4D ratios (Table 5). Moreover, when we analyzed the differences between PWS participants and our healthy controls (Table 5), PWS-GH showed no significant differences with controls in the length of the index and ring finger, but they showed a significantly higher ratio compared to our healthy control sample, both on the right and dominant hands. The significance of these data is difficult to interpret. We could hypothesize that GH treatments during childhood/adolescence bring out the real differences that exist between PWS and healthy controls related to 2D:4D ratios.

Different studies have also shown an association (or lack of association) of several digit measures and ratios with child and adolescent obesity [49] or child body composition [50]. In adult populations, different studies showed that excessive body weight in men and women, and fat accumulation in the upper arms, thighs, and lower legs in women with obesity, are associated with increased prenatal estrogen exposure as measured through 2D:4D [51]. As some authors suggested, this relationship could be of relevance in the field of obesity prevention, as the 2D:4D index (especially of the right hand) may be a useful marker in the early prediction of the elevated risk of developing excessive body weight in humans, although other relevant factors such as eating habits and lifestyle must be taken into account [51].

2D:4D variations have been also studied in psychiatric disorders other than ASD, such as schizophrenia, in different countries and populations, including Spanish populations [52,53,54]. On the other hand, psychotic disorders are as common as ASD in different samples of PWS subjects [8]. The meta-analysis of Laura Fusar-Poli et al. (2021) showed that, upon considering psychiatric disorders individually, significant differences were found in the ASD, ADHD, and addictions groups. In all these disorders, the 2D:4D ratio was significantly lower than in healthy controls. Conversely, the ratios in schizophrenia also showed sexual dimorphism, as the right hand of males with schizophrenia showed higher 2D:4D than healthy controls [55].

Despite the great number of studies and analyses on the relevance of the 2D:4D ratio (and other related ratios) and their relationship to prenatal androgen exposure or MaTtEr, robust evidence for its validity is lacking. On some occasions, studies lack control over important variables. In a recent relevant analysis, Richards and collaborators [49] report the first pre-registered study to investigate 217 mothers’ early pregnancy sex hormone concentrations in relation to their children’s digit ratios measured at 18 to 22 months follow-up. They found that MaTtEr correlated negatively with the right-hand digit ratio (2D:4D) and directional asymmetry (right-minus-left) in the digit ratio (another measure form of the ratio), but when they included demographic and obstetric covariates (child’s sex, maternal polycystic ovary syndrome status, maternal hirsutism score, child’s birth weight, and child’s age at follow-up corrected for gestational age), neither effect remained statistically significant. Finally, they concluded that larger samples are required to determine whether digit ratios are valid proxies for maternal sex hormone exposure [56].

Previous studies showed the relevance of epigenetic regulation of and by SNORD116 and other genes within the locus to the pathogenesis of PWS [1]. Different research and epigenetic approaches could help us to understand the interactions between imprinted genes and metabolism at this locus in PWS. This research not only could help people affected by PWS but also people suffering from other more common metabolic (for example, obesity) and neuropsychiatric human disorders. In addition, there are some future perspectives on the development of CRISPR/Cas9- mediated epigenome editing in the epigenetic therapy of PWS [57]. The knowledge of the influence of the prenatal maternal-testosterone-to-estradiol ratio (MaTtEr) environment on the development of the ‘social brain’ during pregnancy in PWS or ASD individuals could help to develop specific therapies to minimize the cognitive and social responsiveness impact of the illness.

## 5. Limitations and Strengths

Our exploratory analysis has several limitations. The multiple comparison approaches constitute a major limitation concerning the extension of the results.

In addition, our small sample makes subgroup analysis difficult, and the cross-sectional design only allows conclusions about associations. Moreover, the SRS was designed for use in children and adolescents, and the samples used to validate this scale in adults have primarily comprised non-genetically diagnosed individuals; thus, caution is warranted in extending its validity to other populations. Due to the small sample, we did not take into account differences in genetic abnormalities of subgroups. In addition, some studies suggest that the 2D:4D ratio is not a consistent measurement for PT—rMaTtEr [56,58]. Studies on the epigenetics of PWS are relatively scarce [3,59]. Moreover, it was not possible to compare our exploratory results with those of previous studies in PWS. Finally, as some previous authors pointed out, strategies to measure 2D:4D digit ratios could be relevant for the interpretation and comparison of results between samples [47,48].

The main strengths of our data are the originality and the potentially suggestive approach. The study of epigenetic influences in PWS is a relatively unknown topic. Nevertheless, all the limitations of our study must be taken into consideration first.

## 6. Conclusions

Our results could partially meet our preliminary hypothesis. PWS individuals have shorter index finger (2D) and ring finger (4D) lengths than our control group. Moreover, in the global PWS sample, there were no significant differences in the 2D:4D digit ratios compared to our control group. Considering only PWS with previous GH treatment during childhood/adolescence (PWS-GH), index and ring fingers did not show differences in length compared to our control group. However, the PWS-GH showed significantly higher ratios than the controls. One could hypothesize that real differences between controls and PWS emerge only when treating patients with GH during childhood/adolescence.

Finally, as we cited, despite the great number of studies and analyses about the relevance of the 2D:4D ratio (and other related ratios) and their relationship to prenatal androgen exposure or MaTtEr, robust evidence for its validity in the area is lacking. In contrast, the study of epigenetic influences in PWS is a relevant field of future research, and studies on prenatal androgen exposure or MaTtEr could be interesting (but difficult) to approach.

## Data Availability

Not applicable.

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
