# Peer review of "An Exploratory Analysis on the 2D:4D Digit Ratio and Its Relationship with Social Responsiveness in Adults with Prader–Willi Syndrome"

_jcm, 2023, doi:10.3390/jcm12031155_

Round 1

Reviewer 1 Report

My comments to the authors are attached in he "peer-review-2610564.v1.docx" file

Author Response

Article title:

An Exploratory Analysis on Digit ratio 2D:4D and their relationship with Social Responsiveness in Adults with Prader-Willi Syndrome

Dear Reviewer 1:

            Thank you for all your comments and suggestions. We have incorporated all of them in the revised manuscript. Changes to the new manuscript draft are also highlighted in yellow.

            Cordially,

            The Authors

Question 1: "In this paper, Gamez and colleagues describe an initial investigation of two separate, but related issues. First, they suggest that prenatal exposure to elevated testosterone or a high maternal testosterone/estradiol ratio might result in long-term effects on social performance in adults with Prader-Willi syndrome. Secondly, they suggest that the ratio of lengths of the index finger to that of the ring finger ("2D:4D") is a surrogate measure of the maternal testosterone/estradiol ratio. Previous studies, some of which are included in the references, do show that the 2D:4D ratio may be a reflection of prenatal androgen exposure, although as the authors state "robust evidence for its validity is lacking." Although the current study in a relatively small group of adults with Prader-Willi syndrome and a control group did not find a relationship between 2D:4D ratios and social functioning (except for one measure of restricted interests and repetitive behavior), the study was well-planned and the paper is well-written."

Response 1: Thank you very much.

Question 2: "Specific comments: 1. Page 3, lines 116-117: "We hypothesized that people with PWS would have lower 2D:4D ratios than controls…" If the 2D:4D ratio indicates increased prenatal androgen exposure, why would infants with PWS be expected to be exposed to higher androgen levels in utero?"

Response 2: As shown in previous studies, reviews, and metanalyses, higher concentrations of testosterone or higher ratios of testosterone-to-estradiol during prenatal development result in lower 2D:4D ratios [17–19]. PWS is associated with a higher risk for ASD and social responsiveness deficits [16]. Considering psychiatric disorders individually, significant differences were found in the ASD, ADHD, and addictions groups, in which 2D:4D was significantly lower compared to healthy controls [55]. We hypothesize that PWS individuals show the same 2D:4D ratios (or similar) as ASD. We included these sentences there.

References:

  1. Fernández-Lafitte, M.; Cobo, J.; Coronas, R.; Parra, I.; Oliva, J.C.; Àlvarez, A.; Esteba-Castillo, S.; Giménez-Palop, O.; Palao, D.J.; Caixàs, A. Social Responsiveness and Psychosocial Functioning in Adults with Prader–Willi Syndrome. J Clin Med 2022, 11, 1–11, doi:10.3390/jcm11051433.
  2. Lutchmaya, S.; Baron-Cohen, S.; Raggatt, P.; Knickmeyer, R.; Manning, J.T. 2nd to 4th Digit Ratios, Fetal Testosterone and Estradiol. Early Hum Dev 2004, 77, 23–28, doi:10.1016/j.earlhumdev.2003.12.002.
  3. Manning, J.T. Resolving the Role of Prenatal Sex Steroids in the Development of Digit Ratio. Proc Natl Acad Sci U S A 2011, 108, 16143, doi:10.1073/PNAS.1113312108.
  4. Manning, J.T.; Scutt, D.; Wilson, J.; Lewis-Jones, D.I. The Ratio of 2nd to 4th Digit Length: A Predictor of Sperm Numbers and Concentrations of Testosterone, Luteinizing Hormone and Oestrogen. Human Reproduction 1998, 13, 3000–3004, doi:10.1093/humrep/13.11.3000.
  5. Iljin, A.; Antoszewski, B.; Szewczyk, T.; Sitek, A. The 2D:4D Index Is Associated with the Development of Excess Body Weight in Adults, but Not with the Rate of Weight Loss Following Bariatric Surgery. Sci Rep 2022, 12, doi:10.1038/S41598-022-12306-1.
  6. Fusar-Poli, L.; Rodolico, A.; Sturiale, S.; Carotenuto, B.; Natale, A.; Arillotta, D.; Siafis, S.; Signorelli, M.S.; Aguglia, E. Sec-ond-to-Fourth Digit Ratio (2D:4D) in Psychiatric Disorders: A Systematic Review of Case-Control Studies. Clin Psychophar-macol Neurosci 2021, 19, 26–45, doi:10.9758/CPN.2021.19.1.26.

Question 2. "Page 3: In describing the study participants, information regarding previous or current growth hormone treatment should be provided, since growth hormone is likely to affect the hand and finger size. Also information about testosterone and estrogen supplements taken by the PWS participants would be helpful."

Response 2: We reviewed all the medical records of the patients, and we added that information.

Question 3: "3. Page 5, figure 1: This figure is very helpful in illustrating exactly how the digit lengths were measured."

Response 3: Thank you.

Question 4: "4. Page 5, table 1: Height measurements (median and range) are listed for PWS and control groups, but it is incorrect to combine heights for males and females. I suggest either stating median and range of heights for males and females separately, or expressing the heights as SDS (standard deviation scores)."

Response 4: We included two new Tables 1. Table 1a included results only for females and Table 1b results only for males. We included mean and standard deviation, and median plus Inter-quartile range (25-75) for every data.

Question 5: "5. Page 6, table 1: Normal ranges of LH and FSH are different for men and women. Values for the levels in men and women should be listed separately. Also, results for individuals with PWS who were receiving testosterone or estrogen supplements, should not be combined with the results of those patients who were not treated."

Response 5: We reviewed all the medical records of the patients and we do it.

Question 6: "6. Page 7, lines 243 and 248: In view of the assumption that 2D:4D ratio is a measure of prenatal androgen exposure, how do you explain the lack of correlation between the digit ratio and sex of the control group?".

Response 6: It could be related to the specific characteristics of our control sample. 2D:4D ratios are sexually-dimorphic in general populations, but the amount of the differences or the hand involved depends on the concrete population studied.

For example, in a Greek sample (Kyriakidis et al., 2008), men had a lower 2D:4D ratio (0.974 +/- 0.035 for the right hand and 0.973 +/- 0.044 for the left hand) than women (1.002 +/- 0.04 for the right hand and 1.001 +/- 0.045 for the left hand). This difference in the 2D:4D ratio between sexes was statistically significant (p < 0.0001 for the right hand and p < 0.001 for the left hand).

There are also differences related to the age of the participants (Ernstein et al, 2021).

For example, De Sanctis et al. reviewed: "This sexual dimorphism in 2D:4D ratios is apparent by 2 years of age and seems to be established early in life, possibly by the 14th week of gestation. Digits in females attain their maximum length at about 2.2 years (dextral subjects) or 5.1 years (sinistral subjects) earlier than those in males and increase slightly with age. It has also been reported that the 2D:4D ratio is correlated negatively with prenatal testosterone levels".

Bibliography:

Ioannis Kyriakidis, Paraskevi Papaioannidou. Epidemiologic study of the sexually dimorphic second to fourth digit ratio (2D:4D) and other finger ratios in Greek population Coll Antropol. 2008 Dec;32(4):1093-8.

Luisa Ernsten, Lisa M Körner, Martin Heil, Gareth Richards, Nora K Schaal. Investigating the reliability and sex differences of digit lengths, ratios, and hand measures in infants. Sci Rep. 2021 May 26;11(1):10998. doi: 10.1038/s41598-021-89590-w.

Vincenzo de Sanctis, Ashraf T Soliman, Heba Elsedfy, Nada Soliman, Rania Elalaily, Salvatore Di Maio. Is the Second to Fourth Digit Ratio (2D:4D) a Biomarker of Sex-Steroids Activity? Pediatr Endocrinol Rev. 2017 Jun;14(4):378-386. doi: 10.17458/per.vol14.2017.SSE.SexSteroids.

Question 7: "7. Page 7, line 255: "MBI" should be "BMI."

Response 7: We change it.

Question 8: "8. Page 8, lines 266-267: Does the positive correlation of restricted interests and repetitive behavior scale with the 2D:4D ratio in PWS men suggest that this group had decreased exposure to prenatal androgens? Higher prenatal testosterone/estrogen exposure should result in a lower 2D:4D ratio. If there is a positive correlation between restricted interests and 2D:4D ratio, this contradicts the hypothesis stated on page 3, lines 116-117 and should be explained in the discussion section."

Response 8: In our PWS sample, SRS Restricted Interests and Repetitive Behaviour Subscale showed a significant relationship with the digit 2D:4D ratio in the right hand (p = 0.011, r = 0.728) only in men. This could indicate that males tend to exhibit more restricted interests and repetitive behavior depending on their prenatal MaTtEr, but in the opposite direction of our initial hypothesis for the whole PWS sample. We included it in the Discussion.

Question 9: "9. Page 10, line 376: "…there were significant differences in the 2D:4D ratios compared to our control group." But, in the abstract (page 1, line 37) you wrote "…but no significant differences in ratios."

Response 9: Yes, it is a mistake. We changed it.

Thank you,

The Authors

Reviewer 2 Report

This manuscript investigates the relationship between the digit ratio 2D:4D and social responsiveness in adults with Prader-Willi syndrome, a genetic disorder characterized by somatic and behavioral features, and comorbidities, like autism spectrum disorder.

The aim is important for the PWS scientific community in order to get new clues from physical characteristics to better understand the bio-psycho-social underpinning of social behavior in PWS. However, there seem to be some critical concerns about justifying the study and interpreting the findings. Below I address some points that need to be looked at.

Major concerns
Concept and design of the study
Given that the digit ratio 2D:4D (physical level) depends on not only genetic but hormonal environment (hormonal level), and that social behavior (social behavioral level) can be influenced by the hormonal environment (neuroendocrinological level), I believe that the authors should consider more carefully what the current study is going to reveal about the relationship between these three levels.
Specifically, I would like to recommend adding the following analyses that would make the findings more valid.
1. A sex difference of the digit ratio 2D: 4D within the normal control group.
2. A relationship between the hormonal profiles and the digit ratio 2D: 4D in the PWS group.
On the other hand, these additional analyses are likely to make the valuable results more complex. Therefore, I would like to recommend making all the data presentation more sophisticated, as commented below.

Introduction
While the descriptions are enough to cover the comprehensive characteristics of PWS, why such evidence motivates and logically leads to your investigation using the digit ratio 2D:4D, remains poorly understood.
Moreover, there are some overly short paragraphs, which are rather isolated from other paragraphs (L71-73, L107-110).

Methods
The authors need to add a statistics section prior to showing the results, although this study is of exploratory design. This will reduce the redundancy of the descriptions in the discussion section.
The addition of a photo containing a right hand in an individual with PWS may be a good idea, to show the difference between the groups.

Results
These valuable results should be shown as tables or bar graphs in a coherent and non-redundant manner.
The bar graphs with raw data plotting will lend transparency to these analyses.
- “Median (range)” can be concatenated under the tables
- Right hand -> R, left hand -> L
- Signification -> Statistics
- “Male” or “Female” data should be clearly shown in each table.

Conclusion
Please confirm to what extent this description as the significant difference in the 2D:4D ratios is applicable (L375-376).

Minor points

Some typos were found.
This. (line 45), genes among them (line 46), They are (line 186).

Is 2D:4D a formal expression, widely used in such scientific fields?

In summary, the authors should reconsider these points to make the study relevant and show how it can be informative.

Author Response

Article title:

An Exploratory Analysis on Digit ratio 2D:4D and their relationship with Social Responsiveness in Adults with Prader-Willi Syndrome

Dear Reviewer 2:

            Thank you for all your comments and suggestions. We have incorporated all of them in the revised manuscript. Changes to the new manuscript draft are also highlighted in yellow.

            Cordially,

            The Authors

Question 1: "This manuscript investigates the relationship between the digit ratio 2D:4D and social responsiveness in adults with Prader-Willi syndrome, a genetic disorder characterized by somatic and behavioral features, and comorbidities, like autism spectrum disorder.
The aim is important for the PWS scientific community in order to get new clues from physical characteristics to better understand the bio-psycho-social underpinning of social behavior in PWS. However, there seem to be some critical concerns about justifying the study and interpreting the findings. Below I address some points that need to be looked at.
Major concerns: Concept and design of the study: Given that the digit ratio 2D:4D (physical level) depends on not only genetic but hormonal environment (hormonal level), and that social behavior (social behavioral level) can be influenced by the hormonal environment (neuroendocrinological level), I believe that the authors should consider more carefully what the current study is going to reveal about the relationship between these three levels. Specifically, I would like to recommend adding the following analyses that would make the findings more valid:
1. A sex difference of the digit ratio 2D: 4D within the normal control group."

Response 1: We included two new Tables 1. Table 1a included results only for females and Table 1b results only for males. We included mean and standard deviation, and median plus Inter-quartile range (25-75) for every data.

Question 2: "2. A relationship between the hormonal profiles and the digit ratio 2D: 4D in the PWS group. On the other hand, these additional analyses are likely to make the valuable results more complex. Therefore, I would like to recommend making all the data presentation more sophisticated, as commented below."

Response 2: There were no correlations of any 2D:4D ratio with any hormonal laboratory tests (LH, FSH, Testosterone in males, and Estradiol in females). We included this data in the article.

Question 3: "Introduction. While the descriptions are enough to cover the comprehensive characteristics of PWS, why such evidence motivates and logically leads to your investigation using the digit ratio 2D:4D, remains poorly understood."

Response 3: Various previous studies have analyzed the prevalence and relevance of symptoms of autism spectrum disorder (ASD) in individuals with PWS [4,8,12–14]. Our previous study by Fernández-Laffite et al. [16] showed a high prevalence of ASD symptoms.

As we pointed out in the Discussion, the meta-analysis of Laura Fusar-Poli et al. (2021) showed that considering psychiatric disorders individually, significant differences were found in the ASD, ADHD, and addictions groups. In all these disorders, the ratio 2D:4D was significantly lower than in healthy controls. Conversely, the ratios in schizophrenia also showed sexual dimorphism, as the right hand of males with schizophrenia showed higher 2D:4D than healthy controls [55].

Our study is exploratory in its nature, and probably it is related to more questions than answers.

Question 4: "Moreover, there are some overly short paragraphs, which are rather isolated from other paragraphs (L71-73, L107-110)."

Response 4: We modified both.

Question 4: "Methods: The authors need to add a statistics section prior to showing the results, although this study is of exploratory design. This will reduce the redundancy of the descriptions in the discussion section."

Response 4: We included a statistical section.

Question 5: "The addition of a photo containing a right hand in an individual with PWS may be a good idea, to show the difference between the groups."

Response 5: We added it.

Question 6: "Results: These valuable results should be shown as tables or bar graphs in a coherent and non-redundant manner.
The bar graphs with raw data plotting will lend transparency to these analyses.
- “Median (range)” can be concatenated under the tables
- Right hand -> R, left hand -> L
- Signification -> Statistics
- “Male” or “Female” data should be clearly shown in each table."

Response 6: We included bar graphs with raw data plotting of all the analyses. “Median (range)” will be concatenated under the tables. We included Right hand -> R & left hand -> L in Tables. We change the signification for statistics. We make a different Table for males and females (Table 1a & Table 1b).

Question 7: "Conclusion. Please confirm to what extent this description as the significant difference in the 2D:4D ratios is applicable (L375-376)."

Response 7: We changed it. There were no significant differences in the 2D:4D digit ratios for the global PWS sample compared to our control group.

Question 8: "Minor points: Some typos were found. This. (line 45), genes among them (line 46), They are (line 186)."

Response 8: We fixed them.

Question 9: "Is 2D:4D a formal expression, widely used in such scientific fields? In summary, the authors should reconsider these points to make the study relevant and show how it can be informative."

Response 9: Yes, 2D:4D is a formal scientific expression in the area.

For example:

Voracek, M.; Manning, J.T.; Dressler, S.G. Repeatability and Interobserver Error of Digit Ratio (2D:4D) Measurements Made by Experts. Am J Hum Biol 2007, 19, 142–146, doi:10.1002/AJHB.20581.

Richards, G.; Aydin, E.; Tsompanidis, A.; PadaigaitÄ—, E.; Austin, T.; Allison, C.; Holt, R.; Baron-Cohen, S. Digit Ratio (2D:4D) and Maternal Testosterone-to-Estradiol Ratio Measured in Early Pregnancy. Sci Rep 2022, 12, 1–11, doi:10.1038/s41598-022-17247-3.
Richards, G.; Browne, W. v.; Constantinescu, M. Digit Ratio (2D:4D) and Amniotic Testosterone and Estradiol: An Attempted Replication of Lutchmaya et al. (2004). J Dev Orig Health Dis 2021, 12, 859–864, doi:10.1017/S2040174420001294.

Wu, L.; Yao, R.; Zhang, Y.; Wang, Y.; Li, T.; Chen, M.; Liu, W.; Han, H.; Bi, L.; Fu, L. The Association between Digit Ratio (2D:4D) and Overweight or Obesity among Chinese Children and Adolescents: A Cross-Sectional Study. Early Hum Dev 2019, 136, 14–20, doi:10.1016/J.EARLHUMDEV.2019.07.005.

Pruszkowska-Przybylska, P.; Sitek, A.; Rosset, I.; Sobalska-Kwapis, M.; SÅ‚omka, M.; Strapagiel, D.; ŻądziÅ„ska, E. Association of the 2D:4D Digit Ratio with Body Composition among the Polish Children Aged 6-13 Years. Early Hum Dev 2018, 124, 26–32, doi:10.1016/J.EARLHUMDEV.2018.08.001.

Iljin, A.; Antoszewski, B.; Szewczyk, T.; Sitek, A. The 2D:4D Index Is Associated with the Development of Excess Body Weight in Adults, but Not with the Rate of Weight Loss Following Bariatric Surgery. Sci Rep 2022, 12, doi:10.1038/S41598-022-12306-1.

Qian, W.; Huo, Z.; Lu, H.; Sheng, Y.; Geng, Z.; Ma, Z. Digit Ratio (2D:4D) in a Chinese Population with Schizophrenia. Early Hum Dev 2016, 98, 45–48, doi:10.1016/J.EARLHUMDEV.2016.05.003.

The Authors

Round 2
